# Melatonin in Health and Disease: A Perspective for Livestock Production

**DOI:** 10.3390/biom13030490

**Published:** 2023-03-07

**Authors:** Zully E. Contreras-Correa, Riley D. Messman, Rebecca M. Swanson, Caleb O. Lemley

**Affiliations:** Department of Animal and Dairy Sciences, Mississippi State University, Starkville, MS 39762, USA

**Keywords:** antioxidant, circadian rhythms, immune function, cardiovascular effects, endocrinology, skeletal muscle, growth and development, microbiome

## Abstract

Mounting evidence in the literature indicates an important role of endogenous and exogenous melatonin in driving physiological and molecular adaptations in livestock. Melatonin has been extensively studied in seasonally polyestrous animals whereby supplementation studies have been used to adjust circannual rhythms in herds of animals under abnormal photoperiodic conditions. Livestock undergo multiple metabolic and physiological adaptation processes throughout their production cycle which can result in decreased immune response leading to chronic illness, weight loss, or decreased production efficiency; however, melatonin’s antioxidant capacity and immunostimulatory properties could alleviate these effects. The cardiovascular system responds to melatonin and depending on receptor type and localization, melatonin can vasodilate or vasoconstrict several systemic arteries, thereby controlling whole animal nutrient partitioning via vascular resistance. Increased incidences of non-communicable diseases in populations exposed to circadian disruption have uncovered novel pathways of neurohormones, such as melatonin, influence health, and disease. Perturbations in immune function can negatively impact the growth and development of livestock which has been examined following melatonin supplementation. Specifically, melatonin can influence nutrient uptake, circulating nutrient profiles, and endocrine profiles controlling economically important livestock growth and development. This review focuses on the physiological, cellular, and molecular implications of melatonin on the health and disease of domesticated food animals.

## 1. Introduction, Photoperiod, and Synthesis Pathway

Following the discovery of melatonin in 1958 [1], most research efforts focused on understanding the role of melatonin in the reproductive physiology of photosensitive seasonal breeding animals [2,3,4]. Melatonin is known to be elevated at nighttime, which is considered a highly conserved trait among vertebrate animals playing a significant role in circadian rhythms and seasonality of reproduction. For example, the duration of melatonin secretion is different between short days (autumn and winter) and long days (spring and summer) allowing an adaptive response to changes in environmental factors, such as temperature and food availability [5]. Nevertheless, there is existing variability in physiological melatonin concentrations across species (Table 1).

Some domesticated livestock species are short day breeders (sheep, goats, and white-tailed deer) or long day breeders (horses and hamsters) where females experience periods of seasonal anestrus and males experience alterations in testicular weight. However, the domestication of cattle and swine has resulted in the complete loss of seasonal reproduction classifying these species as continuous breeders [5,6]. Multiple studies utilizing ewes and mares have aimed to manipulate light intensity and treat animals with melatonin [4,7,8,9] to advance the breeding season and induce cyclicity. The pharmacological advantage of melatonin advancing the breeding season relies upon the conversion of information from the photic environment indicating the duration of day and night and translating it to a chemical signaling [10]. Throughout the decades, the role of melatonin in seasonal breeding has been extensively reviewed [11,12,13], while recently, other important properties of melatonin in relation to mammals’ health and diseases have become of particular interest. Therefore, the current review article will outline the use of melatonin as a therapeutic to modify antioxidant capacity, cardiovascular function, endocrine metabolism, and immune function in various livestock species.

The photoperiodic stimulation of melatonin synthesis is initiated with the transmission of environmental cues from photoreceptive cells in the retina to the suprachiasmatic nuclei (SCN) in the hypothalamus. The SCN is located dorsal to the optic chiasm and is considered the master biological clock regulating physiological and behavioral processes based on a 24 h cycle allowing organisms to anticipate their physiological needs [14]. Exclusively during the dark cycle, norepinephrine is released from the sympathetic postsynaptic fibers into the pineal gland. The norepinephrine binds to β-adrenergic receptors in the pinealocytes activating the cascade of events that leads to melatonin synthesis [12]. The melatonin precursor, tryptophan, is hydroxylated by tryptophan hydroxylase into 5-hydroxytryptophan and further decarboxylated into serotonin (5-hydroxytryptamine) [15]. Serotonin is increased during the day; nevertheless, during the night there is a 70- to 100-fold increase in the activity of arylalkyl amine N-acetyltransferase (AANAT) which converts serotonin into N-acetyl serotonin (N-acetyl 5-hydroxytryptamine) in the pineal gland [16]. Lastly, hydroxyindole-O-methyltransferase (HIOMT), which has limited circadian fluctuations, converts N-acetyl serotonin into melatonin (N-acetyl 5-methoxytryptamine). Even though AANAT is considered the “melatonin rhythm enzyme” controlling its daily rhythm activity [10], HIOMT activity is the rate-limiting step in melatonin synthesis [17]. Due to melatonin’s lipophilic properties, it is immediately released into the bloodstream after synthesis. Extrapineal tissues and cells such as the retina, stomach, skin, ovaries, and lymphocytes have been shown to have the capacity to synthesize melatonin [18,19,20]; nevertheless, this melatonin does not contribute to the bloodstream pool, rather it is utilized by those same tissues as an antioxidant [21].

Melatonin’s antioxidant capacity is a property of great interest for improving health and preventing diseases in livestock. Interestingly, it has been speculated that melatonin evolved more than 2.5 billion years ago in response to increasing atmospheric oxygen concentrations where bacterial phagocytosis ended in the evolution of organisms containing organelles, such as mitochondria [22]. This group of researchers further suggests that all multicellular organisms can produce melatonin in the mitochondria [22] to defend against reactive oxygen species (ROS). Livestock undergo multiple metabolic and physiological adaptation processes throughout their production cycle. In addition, they can be exposed to unfavorable conditions that lead to excess production of ROS. Melatonin can directly reduce ROS from mitochondrial respiratory chain electron leakage [21] and help to preserve antioxidative enzymes’ functional integrity [23]. This antioxidant property of melatonin deserves further exploration in relation to alterations in the cardiovascular system. Paulis and Simko [24] proposed that melatonin can increase blood flow through two possible mechanisms: (a) indirectly by acting as an antioxidant to decrease vascular oxidative stress or (b) directly by binding to endothelial melatonin receptors. In addition, Pogan et al. [25] proposed that melatonin binds to the melatonin receptor 2 in the endothelial cells, causing an increase in cytosolic Ca^2+^, which activates the nitric oxide synthase and stimulates the nitric oxide production, leading to vasodilation [26]. Thereby, melatonin supplementation to livestock may cause vasoconstriction or vasodilation, altering blood flow and affecting nutrient partitioning depending on the animal’s production status. Outside of melatonin’s actions in the central nervous system, such as binding sites in the SCN, researchers have observed vascular smooth muscle cell binding sites in rodent cerebral and caudal arteries [27]. Earlier reports identified vasodilation properties of supraphysiological concentrations of melatonin in rabbit basilar arteries [28]. However, at lower concentrations melatonin was shown to potentiate norepinephrine-induced vasoconstriction in rat caudal arteries [29]. Interestingly, porcine pulmonary and coronary arteries have been reported to respond to melatonin even though melatonin binding sites were not observed in the vessels [30]. Specifically, melatonin induced a concentration-dependent relaxation of porcine pulmonary arteries, while melatonin potentiated vasoconstriction of both porcine and human coronary arteries. Further studies have proposed that melatonin inhibits nitric oxide signaling in porcine coronary arteries and this could be mediated via melatonin receptor 2 [31].

**Table 1 biomolecules-13-00490-t001:** Summary of melatonin concentrations across specie.

Specie	Sample	Daytime (pg/mL)	Nighttime (pg/mL)	Method	Reference
Human (*Homo sapiens*)	Plasma	1.5–4.9	19.4–42.6	GC-MS	[32]
Swine (*Sus domesticus*)	Plasma	30–40	35–65	RIA	[33]
Swine (*Sus domesticus*)	Serum	42.9	38.4	RIA	[34]
Chicken (*Gallus domesticus*)	Plasma	53	201	RIA	[35]
Chicken (*Gallus domesticus*)	Plasma	67.4–117.9	325–379	RIA	[36]
Sheep (*Ovis aries*)	Serum	10–30	100–300	RIA	[37]
Sheep (*Ovis aries*)	Serum	15.7–36.9	60.7–239.2	ELISA	[38]
Horse (*Equus caballus*)	Plasma	0–10	10–45	RIA	[39]
Horse (Cycling mare in March)	Serum	11.6	13.2	RIA	[40]
Horse (Noncycling mare in March)	Serum	8.9	8.2	RIA	[40]
Cow (*Bos taurus*)	Plasma	5	90	RIA	[41]
Cow (Fall)	Plasma	13.9	27.8	ELISA	[42]
Cow (Summer)	Plasma	12.8	24.73	ELISA	[42]

## 2. Circadian Rhythms, Environmental Factors, and Body Temperature

Melatonin, along with the clock gene network, is known to regulate circadian rhythms in response to environmental factors. Nevertheless, the regulation of circadian rhythms is complex when the SCN, circadian molecular clock, and melatonin are integrated into the mammalian system. Even though multiple peripheral tissues exhibit semiautonomous clocks, the SCN neural circuit is the master circadian pacemaker in mammals [43]. In the SCN neurons, the circadian molecular clock operates based on a transcription and translation oscillatory feedback loop [44], regulating the expression of transcription factors and proteins that control rate-limiting steps in metabolic pathways [45]. Briefly, the CLOCK and BMAL1 proteins heterodimerize (CLOCK:BMAL1) and bind to the E-box enhancer within the nucleus, activating the transcription of the CRY and PER proteins [14] which dimerize in the cytoplasm and create a complex that causes negative feedback in the nuclear CLOCK:BMAL1 repressing its own transcription [46]. Interestingly, in the chicken pineal gland, the CLOCK:BMAL1 heterodimer binds to the E box elements in the AANAT gene promoter, enhancing the transcription of AANAT in a circadian rhythm manner [47]. This implies that the daily rhythm in melatonin production is regulated by the transcription of clock genes and the penultimate enzyme (AANAT) in melatonin synthesis [47].

Circadian rhythms could be altered in response to changes in environmental factors such as photoperiod, stress, food availability, and ambient temperature. Interestingly, it has been reported that melatonin modulates hair growth in various species controlling seasonal molting [48,49]. In vitro, 300 ng/L of melatonin increased growth in cashmere goat hair follicles [48]. The hormonal balance between prolactin and melatonin is essential for seasonal and hair growth modulation. Furthermore, alterations in hair follicle dynamics could result in animals retaining their winter hair coats during the summertime which would be detrimental to these animals, exacerbating the effect of heat stress [50]. 

There is evidence that cattle exhibit a distinguished circadian rhythm in body temperature which fluctuates by season and reproductive stages [51,52]. In humans, the core body temperature reaches its lowest point at 03:00 to 06:00 a.m., completely opposite of the observed increase in melatonin synthesis [53]. These data impelled researchers to investigate the hypothermic properties of melatonin, which could be of great benefit to animal agriculture since livestock are annually exposed to heat stress conditions. Melatonin supplementation was shown to reduce core body temperature during daytime in humans; a drop of 0.30 °C was observed when supplemented with 5 mg of melatonin [54]. More recently, Contreras-Correa et al. [42] reported that 20 mg/d of dietary melatonin decreased vaginal temperature in pregnant heifers during the summertime. Nevertheless, Samuel et al. [55] reported no differences in rectal temperature in late gestational cows after supplementing with melatonin implants during summer. Since the skin is rich in arteriovenous anastomoses, the regulation of blood flow is critical to heat loss and body temperature regulation [53]. Moreover, pregnant animals with increased body temperature exhibit a reduction in uterine blood flow and increased peripheral blood flow, allowing blood distribution to facilitate heat dissipation [56]. Maternal hyperthermia in pregnant mice has been shown to increase embryonic death via an increase in oxidative stress, while melatonin injections alleviated this effect by maintaining a neutral redox status [57]. Cattle grazing endophyte-infected fescue exhibit decreased plasma melatonin [58,59], rough hair coats, and vasoconstriction to peripheral tissues, limiting heat dissipation and increasing body temperature [50,60]. Based on these findings, it is suggested to further investigate the role of melatonin and its usage as a therapeutic in domesticated farm animals experiencing cardiovascular diseases and hyperthermia.

## 3. Immune System

Melatonin is an immunostimulatory, anti-apoptotic, antioxidant, amino acid-derived hormone that modulates immune responses, specifically by enhancing the Th-1 immune pathway [61]. Melatonin’s primary physiological function of regulating the circadian rhythm translates into secondary immune functions including, but not limited to, upregulating cytokine production [62], increasing T cell proliferation [63], stimulating NK cell activity [64], increasing antigen presence [63], and increasing the CD4 to CD8 immune cell ratio [65]. Therapeutic melatonin supplementation in livestock species is a novel area of research, with many publications focused solely on improving reproductive performance. However, within human medicine, melatonin has been shown to have positive effects when treating stress or trauma-induced immunodepression [66,67]. Thus, exploring melatonin’s potential immunotherapeutic role within livestock species is needed, especially related to stress-induced immunosuppression throughout the production cycle. 

In 1986, Maestroni et al. [66,68] inhibited melatonin synthesis in mice by keeping them under constant light or administering β-adrenergic blockers; as a result, these mice exhibited a suppressed immune response when exposed to antigens. This response was characterized by an inability to mount a primary antibody response, decreased immune cells within the thymus and spleen, and a depressed proliferative response of lymphocytes. However, when mice were administered melatonin, all these immunosuppressive effects were reversed [68]. Further exploration of melatonin’s immunostimulatory roles revealed an intricate relationship. Specifically, melatonin modulates the cellular and cytokine profile in both the innate and humoral immune response [69]. 

### 3.1. Innate Immune Response

The innate immune system is the body’s first line of defense. The skin and mucosal membranes form a physical barrier against foreign pathogens. Additionally, chemical environment, microbial competition, enzymatic activity, and movement within the gastrointestinal lumen have similar effects to prevent pathogen invasion. However, if pathogens evade these external defenses, immunocompetent cells begin to produce proteins, cytokines, and chemokines to initiate an inflammatory response [70]. Melatonin can affect cytokine production in immunocompetent cells, resulting in an altered immune response [71]. Specifically, melatonin influences hematopoiesis via neuroendocrine regulation to increase natural killer (NK) and monocyte production within the bone marrow by increasing the production of granulocyte and macrophage cell lineage [72]. Thus, in the presence of melatonin, there is a natural increase in cell’s innate immune components, leading to the immunostimulatory effects observed [69]. 

Melatonin’s antioxidant properties are hypothesized to aid in macrophage phagocytic activity [73]. The microbiocidal properties of macrophages are associated with excessive nitric oxide (NO) production that can be harmful to the body [74]. However, melatonin suppresses NO synthase, leading to decreased NO concentrations in macrophages and increased phagocytic activity [75]. Melatonin’s influence on immune cell proliferation and efficacy explains part of the immunostimulatory phenomenon observed, but melatonin’s ability to alter cytokine proliferation and control adaptive immune response is equally important. 

### 3.2. Adaptive Immune Response

Melatonin influences immunocompetent cells to shift cytokine production to increase IL-2, IFN-gamma, and IL-6 in CD4 cells [76]. In monocytes, melatonin increases IL-1, IL-6, TNF-alpha, and IL-12 production [77]. An immune response is typically dictated by the T cell that is activated and cytokines produced. Th-1 responses are activated to target intracellular pathogens via a pro-inflammatory response [66]; this type of response is extremely effective against pathogens but can lead to uncontrolled tissue damage [78]. Th-2 responses balance the Th-1 response by producing interleukins to increase eosinophilic cells and anti-inflammatory responses [78]. Thus, melatonin’s ability to shift cytokine production can drastically alter the Th-1 and Th-2 balance of the immune response. 

Melatonin stimulates Th cells to secrete opioid peptides [62]. Nelson and Drazen [79] hypothesized that this function of melatonin is derived from physiological adaptations that must occur during winter months. Thus, as endogenous melatonin synthesis increases during the winter months, the immunostimulatory effects of melatonin allow the organism to better cope with physiological stressors [79]. Based on the cytokine profile, melatonin favors the pro-inflammatory Th-1 immune response and decreased melatonin is correlated with impaired Th-1 immune responses [80]. Together, there is clear evidence that melatonin is a crucial component of a normally functioning immune system but the implications within livestock species remain unclear and more research is needed for further elucidation.

### 3.3. Immunostimulatory Melatonin in Livestock Production

Within livestock production, there are several unavoidable stressors animals will encounter throughout the production lifecycle. Some examples of these stressors include transportation stress, handling stress, environmental stress, herd dynamics, and sickness (Figure 1). An increased concern with stressed animals is a decreased immune response leading to chronic illness, weight loss, or decreased production efficiency, all of which translate to economic loss. However, melatonin’s immunostimulatory properties could alleviate the effects that stressors cause within livestock species. 

In pregnant sheep, melatonin has been used as a vaccine adjuvant; researchers found that melatonin administration significantly improved the immune responses to the vaccine antigen [81]. Improving vaccine immune response and antibody production in prepartum females can increase colostrum quality, resulting in healthier offspring; thus, this potential use should be further explored. Interestingly, ewes implanted 40 days before lambing with 18 or 36 mg of melatonin exhibited increased IgG concentrations in the colostrum and decreased somatic cell count in the subsequent lactation compared to nonimplanted ewes [82]. In dairy cows, melatonin has been extensively studied to be used as a therapeutic to increase reproductive success in cows experiencing heat stress [83,84]. Generally, melatonin has been successful in reducing the number of days open and the repeat breeding syndrome within heat-stressed dairy cows [83]. However, livestock studies using melatonin as a therapeutic during routine stressors are limited. Based on human and rodent research, melatonin has substantial potential to mitigate stress-induced immunodepression [62,67]. Thus, future research evaluating melatonin supplementation during routine stressors (transportation, handling, weaning, etc.) is needed to truly understand the positive impacts melatonin could have on livestock production efficiency. 

## 4. Microbiome

Immune system modulations have specific and often intricate relationships with microbial fluctuations throughout the body. These relationships have only begun to be explored in both human and livestock species. The progression in sequencing depth, metabolomic analysis, and bioinformatics allows researchers to understand not only which bacteria are present, but also their metabolic capacity and roles within the biome. Thus, exploring the relationship between animal immune status and microbial presence is an emerging field within livestock. Melatonin is a known immunomodulator, but the extent to which melatonin concentrations impact microbial populations throughout the body is quite impressive. 

### 4.1. Gut Microbiome

Through immune-modulatory mechanisms, melatonin has been shown to improve microbial dysbiosis in humans [85]. Specifically, melatonin works through toll-like receptor (TLR) 4, which is responsible for pathogen-associated molecular pattern (PAMP) signaling primarily involving lipopolysaccharide (LPS) on gram-negative bacteria [86]. Interestingly, melatonin demonstrated rhythmic concentrations within the GI tract of mice that are 400-fold greater than the pineal gland [87], which is reflective of the high expression of melatonin receptors and enzymes for melatonin production [88]. Moreover, studies have shown that gut microbes exhibit circadian rhythms and patterns similar to the host that can affect microbes’ relative abundance, absolute abundance, and metabolomic function [89]. Melatonin regulates the biological clock in the host [90]; thus, it is clear the microbial circadian rhythm and function of the gut microbes are tied to melatonin. Together, this evidence presents a strong case for melatonin to cause physiological changes within the digestive tract. 

The identification of rhythmicity of the ruminant gut microbiome warrants further research, but interestingly melatonin’s role within the gut microbiome could be linked to salivary origins. Salivary melatonin has roles in regulating inflammatory processes, promoting antioxidant responses, and rapid healing within oral wounds [91]. Moreover, salivary melatonin concentrations follow a similar circadian rhythm to the pattern in ruminal fluid and ruminal muscularis [92]. Thus, melatonin secreted into the saliva could be impacting microbial communities throughout the gastrointestinal tract via circadian fluctuations. A study within lactating Holstein cows (n = 6) demonstrated a circadian rhythm within the rumen gut microbial populations and found that microbe relative abundance changed with ruminal melatonin concentrations. Specifically, increased melatonin concentrations resulted in increased relative abundance of the families Preovotellaceae and Muribaculaceae; there was a decrease in the relative abundance of the families Succininivibrionaceae and Veillonellaceae [92]. This is concurrent with previous research demonstrating melatonin’s ability to negatively affect gram-negative bacteria via cytokine production and altered metabolism [93]. Based on the results, Ouyang et al. [92] hypothesized that the oscillation of melatonin concentrations within the gastrointestinal tract alters key metabolic pathways that impact the dominant phyla (Firmicutes, Proteobacteria, and Bacteroidetes) within the rumen. 

### 4.2. Reproductive Tract Microbiome

There is limited literature on livestock evaluating the relationship between dietary melatonin and the reproductive tract microbiome. However, within the singular published study, 60 days of dietary melatonin supplementation altered the beta diversity of the vaginal tract microbiome [94]. The authors contributed this observation to melatonin’s role in altering uterine artery blood perfusion and potentially oxygen perfusion to the tissue but, given melatonin’s innumerable roles within the immune system, there could be some inadvertent immune responses within the reproductive tract resulting in compositional changes [94]. 

### 4.3. Melatonin in Livestock Microbiomes

Taken together, melatonin cyclicity obviously impacts the microbiomes within the host. In humans, melatonin’s therapeutic role in treating dysbiosis-associated conditions such as inflammatory bowel disease, chronodisruption-induced dysbiosis, obesity, and neurophsychiatric disorders is being explored [95]. Thus, the limited literature investigating therapeutic melatonin in livestock to decrease microbiome dysbiosis and increase overall efficiency is problematic; future research must focus on the secondary effects of melatonin supplementation specifically related to immune and microbial modulations and how these effects can be harnessed to increase overall production efficiency.

## 5. Skeletal Muscle and Growth and Development

There has been recent progress made in our understanding of melatonin’s effects on function, growth and development, and therapeutic benefits in diseases and dysfunction of skeletal muscles. In C2C12 mouse myoblasts, 0.5 mM and 1 mM of melatonin increased proliferation rates from 48 to 96 h [96]. While 0.5 mM of melatonin did not affect the transcript abundance of myogenic regulatory factors, 1 mM and 2 mM of melatonin decreased myogenin (MyoG) and embryonic myosin heavy chain (eMyHC) [96]. Furthermore, 2 mM of melatonin reduced the transcript abundance of all fusion factors assessed, while 1 mM of melatonin only reduced some fusion factors [96]. Interestingly, 2 mM of melatonin increased the rate of apoptosis [96]. These data indicate melatonin can promote the proliferation of skeletal muscle cells, inhibit differentiation and fusion, and increase apoptosis in a dose-dependent manner. In L6 mice myotubes treated with 100 ng/mL TNFα in culture, 100 nM of melatonin improved cell viability and reduced markers of apoptosis, including p38-MAPK, JNK, and cleaved caspase-3 [97]. Early and old-aged mice receiving 10 mg/kg/d of melatonin in their feed had increased muscle weight, body weight, and muscle-to-body weight ratio [98]. Furthermore, these mice had reduced internal damage, collagenous tissue accumulation, and nuclei apoptosis in skeletal muscle fibers of the gastrocnemius muscle [98]. Young, early, and old-aged mice receiving melatonin in this study had increased whole-body anaerobic respiration, evidenced by increased lactate production [98]. Hindlimb skeletal muscle blood flow was increased in mice receiving 100 mg/kd/d of melatonin in their drinking water [99]. Furthermore, these melatonin-treated mice had improved insulin sensitivity and glucose utilization [99]. Similarly, rats receiving 0.5 mg/kg/d of melatonin in their drinking water also had improved insulin sensitivity and glucose utilization [100]. Blood glucose rhythmicity was diminished, and blood glucose concentrations were decreased during daylight hours among melatonin receptor 1 and melatonin receptor 2 knockout mice [101]. These data indicate that melatonin can improve skeletal muscle function, including glucose homeostasis and metabolism, which can improve overall health and well-being in people with metabolic diseases or the elderly. Similarly, these data allow for speculation that melatonin can improve growth and health and wellbeing in livestock destined to become protein sources for human consumption. Melatonin implants releasing 2 mg/kg/d in goats had no effect on carcass weight, dressing percentage, *longissimus dorsi* cross-sectional area, essential amino acids, total amino acids, or individual amino acids [102]. Additionally, there were no differences in muscle pH, muscle water content, or meat color in the *longissimus dorsi*, *biceps femoris*, or *gluteus* muscles in goats implanted with melatonin [102]. Furthermore, goats that received implants in June had decreased protein content in all three muscles but only decreased ether extract content in the *gluteus* muscle [102]. Interestingly, whole-muscle and whole-body growth has been variable in melatonin studies; however, the therapeutic benefits at the cellular level in skeletal muscle would likely reduce adverse health issues and create a beneficial environment for skeletal muscle growth. Further research should investigate the efficacy of melatonin implants to better understand release rates and its ability to enter circulation from subcutaneous spaces. While significant gains have been made in our understanding of melatonin’s role in skeletal muscle function, there is a need for more data; specifically, evaluating melatonin’s impact on growth and development in skeletal muscle. These studies allow us to speculate that melatonin could improve poor growth and development, although it is likely dose dependent.

## 6. Amino Acids in Livestock Maternal Blood

Since melatonin is synthesized from tryptophan there is increasing interest in the role of melatonin on circulating amino acids. There were no effects of melatonin on total amino acid concentrations in the saphenous artery or uterine vein in mid-gestation nutrient-restricted ewes receiving 5 mg of dietary melatonin daily [103]. Similarly, there were no effects of melatonin on total branched-chain amino acids in these sheep [103]. Melatonin rescued the effects of nutrient restriction in total amino acids but not essential amino acids in late gestation cows receiving 20 mg of dietary melatonin daily in the fall [104]. In these same cows, amino acids were evaluated by a transport system in which melatonin also rescued the effects of nutrient restriction in System A, System N, and Anion amino acids [104]. Furthermore, fall-supplemented melatonin exhibited similar rescue effects in individual amino acids including valine, α-aminobutyric acid, aspartic acid, glutamic acid, α-aminoadipic, acid lysine, tyrosine, and cystine in these cows [104]. Interestingly, in another replicate in which cows received melatonin in the summer, melatonin rescued the effects of nutrient restriction in System Br and System Bo amino acids [104]. However, there were no effects of melatonin in total amino acids, essential amino acids, or individual amino acids in summer-supplemented cows [104]. These data indicate that melatonin may be an effective therapeutic for nutrient stress during gestation when considering altered circulating amino acids. Furthermore, melatonin may be a more effective therapeutic in certain seasons. Seasonality research using melatonin is limited and should be expanded upon to better understand the efficacy of melatonin as a therapeutic. Interestingly, the use of melatonin as a therapeutic has quickly become a topic of research in several diseases, while studies into its effect on amino acids and other metabolites remain lesser. In a mouse breast cancer model, 40 mg/kg of body weight melatonin injections rescued the effects of breast cancer on circulating amino acids [105]. Specifically, melatonin reduced tryptophan, proline, ornithine, methionine, lysine, isoleucine, glutamate, and citrulline while increasing aspartate, leucine, lysine, proline, serine, and valine in breast cancer bearing mice [105]. These alterations in amino acids due to melatonin supplementation were similar to concentrations in the control mice [105]. These data suggest melatonin may be effective in regulating amino acids in breast cancer patients, which could allow for reduced tumor growth by lessening the fuel source for cancer cells. Studies evaluating melatonin as a therapeutic should consider investigating amino acids, as it may show improvement when nutrients are inadequate or overly abundant during pregnancy or disease.

## 7. Endocrine and Receptor Pathways

The amplitude of melatonin secretion has been associated with steroid and prostaglandin metabolism in rats and sheep. Progesterone production is stimulated in luteal cell cultures treated with melatonin [106], while melatonin supplementation decreased prostaglandin F2 and E2 in endometrial and hypothalamic cultures [107,108]. In addition, melatonin treatment in rats reduced uterine estrogen receptors and increased uterine progesterone receptors, while concomitantly reducing uterine contractile response to oxytocin compared with the controls [109]. In human breast cancer cell lines, melatonin interacts with estrogen receptors as a selective estrogen receptor modulator, and it has been implicated in reducing estrogen synthesis in steroidogenic tissues [110]. Along those same lines, melatonin reduced the activity and expression of aromatase, responsible for synthesis, and sulfatase, responsible for the bioavailability of estrogens. This increased activity of estrogen sulfotransferase generates an estrogen sulfate with low biological activity and a long half-life [110]. 

In pregnant cattle, dietary melatonin supplementation during the third trimester of pregnancy decreased both estradiol-17beta and progesterone concentrations [111]. This decrease in steroid concentrations could be related to metabolism, as treatment with physiological concentrations of melatonin increased the enzymatic activity of cytochrome P450 1A. This enzyme participates in a number of metabolic pathways, including the conversion of estradiol to 2-hydroxyestradiol metabolite, which can be further metabolized to 2-methoxyestradiol via the catechol-O-methyltransferase enzyme [112,113]. Interestingly, deficiency in 2-methoxyestradiol production has been associated with pre-eclampsia-like phenotypes in mice [114]; therefore, alterations in estradiol metabolism after melatonin exposure could alter uteroplacental development during pregnancy. This melatonin-mediated response may be related to the activation of the aryl hydrocarbon receptor, which binds indole-containing chemicals directly, thereby increasing the expression of cytochrome P450 enzymes [115]. However, receptor-mediated pathways cannot be ruled out. In bovine endometrial epithelial cells treated with increasing concentrations of estradiol, we observed decreased melatonin receptor 1 expression, while treatment with progesterone increased melatonin receptor 1 expression [111]. These results are important because uteroplacental steroid and prostaglandin synthesis and metabolism are associated with nutrient transport capacity and uterine blood flow [116,117]. In addition, estrogen has been implicated in blocking adrenergic uterine arterial tone [118] and elevated melatonin may be implicated in decreasing estrogen concentrations or estrogen sensitivity, which has direct implications for controlling uterine blood flow during compromised pregnancies.

## 8. Conclusions and Future Directions

In summary, melatonin’s properties have been observed to impact cardiovascular, immune function, growth and development, and endocrine pathways in livestock species. Altering antioxidant capacity contributes to melatonin-mediated physiological changes, while melatonin receptor-mediated pathways have been proposed in sheep and cattle. Disrupting photoperiod and altering endogenous melatonin secretion in livestock can have profound effects on cardiovascular function, core body temperature, immune health, and growth, which are all major components of the animal agriculture industry. Previous research has focused on implicating melatonin in regulating reproductive performance in livestock species; however, this review has shed light on innovative pathways that need to be targeted in animal agriculture. Specifically, understanding how melatonin or circadian disruption can impact the microbiome of economically important livestock could lead to significant strategies to decrease morbidity and mortality during livestock production. Furthermore, recent evidence linking melatonin to alterations in systemic metabolites, such as amino acid concentrations and steroid hormone profiles, elucidates novel mechanisms which can be harnessed to improve the efficient growth and development of livestock species. 

## Figures and Tables

**Figure 1 biomolecules-13-00490-f001:**
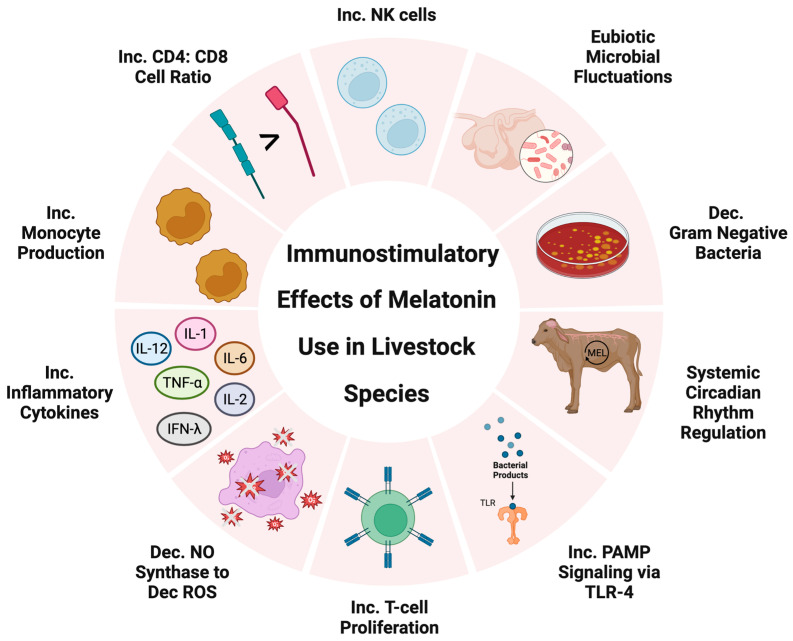
Figure depicting how peripheral concentrations of melatonin impact the immune responses throughout the body via systemic circadian rhythm regulation. Increased concentrations of melatonin altered eubitoic microbial populations, increased immunocompetent cells (natural killer cells (NK), monocytes, and T cells), and cytokine production (IL-1, IL-2, IL-6, IL12, TNF-alpha, and IFN-gamma), decreased nitric oxide (NO) synthase production, resulting in decreased reactive oxygen species (ROS), and increased pathogen-associated molecular patterns (PAMP) signaling via toll-like receptor 4 (TLR-4). The figure was created with BioRender.com (accessed on 29 January 2023).

## Data Availability

Not applicable.

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
