# Peer review of "Melatonin in Health and Disease: A Perspective for Livestock Production"

_biomolecules, 2023, doi:10.3390/biom13030490_

Round 1
Reviewer 1 Report
Congratulations on writing this paper.
The standard structure of the article was well complied and the writing was acceptable; however, I suggest minor revisions.
It would be better to insert citations after the sentence is finished, for example in cases of mentioning the researchers name, it's more suitable to cite at the end of the sentence.
In part 2 (Circadian rhythms et al.) There was a sentence that AANAT gene was mentioned in lowercase. Writing in uppercase form would be a nice change. At the same section, the time of reaching at lowest concentration of melatonin should be written in standard format.(03:00 AM to 06:00 AM)
In Adaptive Immune Response section, I recommend to change the last sentence from an interrogative sentence to a declarative one.
Except the notes mentioned above, other parts of the paper was good and there were no errors that need to be noted. Generally, the article was well written and the data mentioned in the paper was useful.
Author Response
Congratulations on writing this paper.
The standard structure of the article was well complied and the writing was acceptable; however, I suggest minor revisions.
It would be better to insert citations after the sentence is finished, for example in cases of mentioning the researchers name, it's more suitable to cite at the end of the sentence.
In part 2 (Circadian rhythms et al.) There was a sentence that AANAT gene was mentioned in lowercase. Writing in uppercase form would be a nice change. At the same section, the time of reaching at lowest concentration of melatonin should be written in standard format.(03:00 AM to 06:00 AM).
Author: Corrected
In Adaptive Immune Response section, I recommend to change the last sentence from an interrogative sentence to a declarative one.
Author: Corrected
Except the notes mentioned above, other parts of the paper was good and there were no errors that need to be noted. Generally, the article was well written and the data mentioned in the paper was useful.
Reviewer 2 Report
The manuscript entitled ‘Melatonin in health and disease: A perspective for livestock production’ is an interesting review that focuses on physiological, cellular, and molecular implications of melatonin on health and disease of domestic animals, apart from the control of reproduction in seasonal breeders.
However, the authors may consider some references that have not been included but fit well the objectives of this review i.e. for colostrum quality and milk yield the references https://doi.org/10.3390/ani12101257 and https://doi.org/10.3390/ani11113161, for heat stress in ewes the reference https://doi.org/10.3390/antiox9030266
Concerning the livestock production written in the title of the manuscript, the authors should also review the impact of internal or exogenous melatonin on milk traits in milk producing animals (e.g. https://doi.org/10.1080/1828051X.2021.1904796; https://doi.org/10.3390/ani10101721; https://doi.org/10.3390/ani12101257) and on newborns’ survival (e.g. https://doi.org/10.1093/jas/skaa344; https://doi.org/10.1093/jas/skaa372)
Author Response
The manuscript entitled ‘Melatonin in health and disease: A perspective for livestock production’ is an interesting review that focuses on physiological, cellular, and molecular implications of melatonin on health and disease of domestic animals, apart from the control of reproduction in seasonal breeders.
However, the authors may consider some references that have not been included but fit well the objectives of this review i.e. for colostrum quality and milk yield the references https://doi.org/10.3390/ani12101257 and https://doi.org/10.3390/ani11113161, for heat stress in ewes the reference https://doi.org/10.3390/antiox9030266
Author: The authors appreciate these suggestions. The citation for the article from Canton et al., 2022 was included to the “Immunostimulatory Melatonin in Livestock Production” section due to its relevancy in colostrum quality.
Concerning the livestock production written in the title of the manuscript, the authors should also review the impact of internal or exogenous melatonin on milk traits in milk producing animals (e.g. https://doi.org/10.1080/1828051X.2021.1904796; https://doi.org/10.3390/ani10101721; https://doi.org/10.3390/ani12101257) and on newborns’ survival (e.g. https://doi.org/10.1093/jas/skaa344; https://doi.org/10.1093/jas/skaa372)
Author: The authors appreciate these suggestions, but we would like to focus this review article more on the implications of melatonin in health and disease. We will keep citations for future refences.
Reviewer 3 Report
The manuscript is well-written and of interest.
Author Response
The manuscript is well-written and of interest.
Author: We appreciate the reviewer's comment
Reviewer 4 Report
This paper should supplement the relevant research of melatonin in the field of animal breeding. Especially in the short sunshine animal sheep
Author Response
This paper should supplement the relevant research of melatonin in the field of animal breeding. Especially in the short sunshine animal sheep
Author: We appreciate the reviewer's comment.
Reviewer 5 Report
This manuscript precisely summarizes roles of melatonin in livestock with a lot of papers.
-Animal species discussed are appropriate.
-Properties of melatonin in relation to health and disease of livestock are interesting subjects.
-Effect of melatonin on gut microbiome and rumen microbiome are discussed respectively.
-Effect of melatonin on beef cattle are not described. Differences in the effect between fattening and finishing state are more interesting.
-Melatonin improves insulin sensitivity and influences estradiol metabolism. These phenomenon suggest application of melatonin as supplement to increase productivity of livestock.
Author Response
This manuscript precisely summarizes roles of melatonin in livestock with a lot of papers.
-Animal species discussed are appropriate.
-Properties of melatonin in relation to health and disease of livestock are interesting subjects.
-Effect of melatonin on gut microbiome and rumen microbiome are discussed respectively.
-Effect of melatonin on beef cattle are not described. Differences in the effect between fattening and finishing state are more interesting.
Author: Literature involving melatonin supplementation and beef cattle fattening is scarce.
-Melatonin improves insulin sensitivity and influences estradiol metabolism. These phenomenon suggest application of melatonin as supplement to increase productivity of livestock.
Round 2
Reviewer 4 Report
Accept in present form